# Exploring Characteristics of Regenerative Business Models through a Delphi-Inspired Approach

**Linda Drupsteen \* and Ingrid Wakkee**

Centre for Economic Transformation, Amsterdam University of Applied Sciences,
1102 CR Amsterdam, The Netherlands; i.a.m.wakkee@hva.nl
\* Correspondence: l.drupsteen@hva.nl; Tel.: +31-643-611-601

**Abstract:** Amidst escalating environmental and social challenges, this study explores regenerative business models' definition and characteristics. While sustainable models have made considerable strides in research, policy, and practice, the advent of regenerative business models offers a progressive leap forward. Regenerative business models aspire to contribute to ecological restoration and societal well-being. The regenerative business model concept is, however, still in its infancy and lacks a comprehensive definition. Our study aims to expand this knowledge, using a Delphi-inspired approach that builds on the knowledge of academic and business experts. Our approach includes three rounds of surveys: an open-ended survey, a survey for rating and ranking the earlier responses of all participants, and a final survey to select key characteristics. We investigate patterns and distinctions among regenerative, regenerative business, and regenerative business models, and analyze their positioning vis-a-vis circular and net-positive models. Findings underscore that organizations adopting regenerative business models focus on planetary health and societal well-being. They generate value across multiple stakeholder levels, including nature, societies, customers, suppliers, shareholders, and employees. Despite overlapping with circular and net-positive models, regenerative business models also emphasize interdependencies between humans and nature, and provide a more holistic approach, centered on restoration rather than mere mitigation.

**Keywords:** regenerative business; Delphi study; business models; net-positive; entrepreneurship

## 1. Introduction

In recent years, there has been growing emphasis on sustainable business practices, driven by the pressing environmental and social challenges faced by both society and businesses. As the state of our planet and societies continues to deteriorate, it is critical for entrepreneurs to reassess the way in which they create value for society. They need to find ways to reduce pollution and be fairer to reduce inequality and thus reassess their sustainable business practices. This will require innovation both in terms of the products and services they bring to market, but also in the business models they use to create and deliver value.

Sustainable business models have already gained significant ground in research, policy, and practice. For instance, much attention has been paid to circular models that aim to limit negative impact on the environment, by minimizing waste and maximizing resource efficiency [1], and net-positive models which strive for positive impact to the environment, for instance by compensating for negative effects [2]. However, despite the attention for these sustainable models, the recent emergence of regenerative business models presents an opportunity to take a step further. Regenerative business models are a particular type of sustainable business models that aim to contribute to ecological restoration and societal well-being. This concept of the regenerative business model is, however, still in its infancy and lacks a comprehensive definition [3]. For instance, some scholars argue that regenerative business models are primarily concerned with ecological restoration and

resource regeneration, echoing the principles of circular economy [4,5]. Most studies focus, however, on the socio-ecological approach, including also the regeneration of communities and including human well-being as part of the system [6–8].

In seeking deeper insights into the core design principles of regenerative business models, we observed that few studies have been conducted on the topic and that most insights can be derived from anecdotal evidence [9]. This anecdotal evidence suggests that regenerative business models emphasize the relational nature of human and natural systems with the intention to restore both nature (including climate and biodiversity) and (indigenous) communities, while honoring and leveraging diverse cultural heritages [10]. Consequently, value propositions of regenerative business models encompass a broad set of values, offered both to the customer and the environment. Despite these insights, it remains undisputed that there is little existing research and that there is a clear need to collaboratively establish a definition and key characteristics of regenerative business models. Our study seeks to fill this void by engaging experts from diverse backgrounds to collectively develop a nuanced understanding of this emerging business paradigm.

The main objective of this study is to contribute to the knowledge on regenerative business models through identification of key elements and characteristics. Thus, there are two main questions guiding our research. First, how can we define and characterize the concept of regenerative business models? Second, how are such business models distinctive from other sustainable business models?

To answer these research questions, after reviewing previous conceptual and empirical studies on regenerative business (models), we employed a Delphi-inspired approach that integrates the perspectives of academic experts, business representatives, and professionals from the broader business ecosystem. Drawing inspiration from the Modified Delphi Method [11], this research follows a seven-step process to solicit opinions from experts and have them rank these opinions. Through this iterative approach, we aim to distill collective wisdom, and establish a comprehensive definition and key characteristics of regenerative business models. Specifically, we examine key patterns and distinctions identified by the experts when discussing regenerative, regenerative business, and regenerative business models. Additionally, we explore how these concepts can be positioned vis-a-vis circular and net-positive business models. Comparing and contrasting regenerative business models with circular and net-positive models is crucial for understanding their distinct features and identifying potential synergies, thereby guiding businesses in informed decision-making for sustainable practices.

In doing so, our study makes several contributions to the fields of sustainability, entrepreneurship, and business models. First, it addresses a significant research gap by defining and characterizing regenerative business models and comparing these with other sustainable models, specifically circular and net-positive business models. Second, our study underscores the importance of a holistic approach and of systems thinking in the context of regenerative business models. We emphasize the interconnectedness of ecological, social, and economic aspects, highlighting the need for entrepreneurs to consider their impacts and dependencies on natural systems, communities, and society. Our findings will not only inform academic discourse, but will also provide practical insights for businesses looking to adopt regenerative practices.

The remainder of this paper is organized as follows. The following section presents a brief overview of the literature on regenerative business models. Next, we will describe the methodology used for data collection and analysis. Subsequently, we present our findings, and the paper concludes with a discussion of implications, limitations, and needs for further research.

## 2. Theoretical Background

Given the current condition of our planet and society, entrepreneurs are compelled to reassess their approach to creating societal value, placing a significant emphasis on sustainability and equity. This reevaluation extends beyond product and service innovation

to encompass a fundamental shift in the methodologies employed for value creation and delivery. This transformation has become a focal point in scholarly discourse, notably within the realm of sustainable business models [12–14]. Broadly speaking, these models aspire to provide a comprehensive description of value capture, integrating economic, environmental, and social dimensions [15–17].

Moreover, there has been an expansion of business models aligned with Sustainable Development Goals (SDGs), addressing challenges such as inclusive growth [18,19]. Notably, the recent literature reflects a discernible shift from viewing sustainability as a secondary objective subordinate to economic profit, as exemplified in the win-win argument rooted in financial motivation, towards a perspective where sustainability assumes a central role in 'strong sustainability' business models [20,21].

With the rise of such sustainability-oriented business models, tools for the design of business models to describe value capture have been extended and adapted to multiple types of value. For instance, Ref. [22] extended the traditional business model canvas [23,24] by adding environmental and social layers, emphasizing a holistic approach. Another tool that integrates these shared values, but which has gained less ground in the literature, is the "flourishing business canvas" (https://flourishingbusiness.org/ accessed on 4 February 2024).

Despite these advancements, there are still notable gaps in our understanding of sustainable business models. Developing business models for sustainability purposes has traditionally taken a neutral approach, primarily aimed at avoiding further damage without actively attending to the restoration of the damage already done [16]. However, there is a growing recognition of the need to move beyond mere sustainability and towards business models that actively seek to restore and create thriving living systems [25]. In this study, we hone in on a specific emerging category of sustainable business models known as regenerative business models.

The rise of regenerative business models introduces an additional layer to sustainable practices by aiming to contribute not only to sustainability, but also to ecological restoration and societal well-being. This nascent concept, lacking a comprehensive definition, signifies a crucial area where research efforts should be directed to fill the existing gaps and gain a more nuanced understanding.

Regrettably, the scientific literature on regenerative business models is currently limited. A recent Scopus search for 'regenerative business' in the fall of 2022 yielded only a handful of results, with few directly addressing the concept of regenerative business models. Some studies touched upon the idea of regenerative business practices, but failed to provide detailed definitions or implementation strategies. Upon proper inspection, one paper [26] that was originally included, actually focused on business models for regenerative medicine and was therefore removed from the list. Similarly, Ref. [27] provided an overview of developments anticipated in the agribusiness sector in the aftermath of the COVID-19 pandemic. However, even though the rise of regenerative business is mentioned, due to the absence of theoretical elaboration or empirical investigation we also excluded this paper from further analysis.

Amongst the remaining studies, Ref. [28] briefly touched upon the concept of regenerative business practices within the context of postpartum care resorts. They highlighted how regenerative practices might benefit customers, the resort, and the community through the creation of collaborative networks promoting holistic wellness and environmental sustainability. Yet, they do not elaborate on what this might entail or how this could be implemented.

Likewise, another case study by [29] focuses on a chemical manufacturing innovation (SMDR) as a potential platform to speed up business model transformation within the chemical sector. While the authors mention aspects such as enabling local, flexible, and resilient manufacturing (p. 2), regenerative approaches to fine-chemical production, regenerative supply chains, and a regenerative system that minimizes the loss of biological and

technical nutrients (p. 8), they do not define or explain these attributes or the mechanisms by which they can be achieved.

In contrast, Ref. [7] does not specifically present examples of regenerative business models, but merely addresses the need to move towards a systems-based perspective. They posit that a company should consider social and environmental concerns beyond financial aspects in their activities and interactions with stakeholders. Resulting from their study is a scale for regenerative business strategies (from restoration to preservation and to enhancement), emphasizing the shift from a business-centered logic to a systems-based approach to support social-ecological systems [30].

Additionally, Ref. [4] underscores the urgency of businesses to transition from "doing less bad" to "doing more good". This study specifically focuses on regenerative business practices in small- and medium-sized enterprises (SMEs). The study explores insights from Australian SME case studies that focus on restoring planetary systems and adopting innovative, nature-centric approaches, emphasizing the importance of advocating for ecosystem regeneration within the "Action Framework for Regenerative Business".

Notably, Ref. [8] stands out as one of the few authors who explicitly defines regenerative business as "the process in which businesses innovate to continue and stimulate the vitality of the eco-system toward and beyond value creation for humans and nature" (p. 261). For Hofstra, nature-based solutions form the basis of economic transformation, but he also stresses the importance of spirituality for business. Focusing particularly on eco-spirituality, Hofstra argues that the "inherent sacredness of nature is often denied, even within the environmental movement and advanced environmentally friendly businesses, but it can be found in traditional beliefs and cultures" (p. 267). Yet, he also posits that ecological and economic forces might actually reinforce one another, and can even form the basis for creative destruction as instigated by entrepreneurs.

Ref. [31] presents a personal reflection on the economic, environmental, and social systems relevant to the mining industry, emphasizing the need for regenerative business models in mine facility regeneration. The paper highlights the complexities of environmental systems and discusses a specific land regeneration project in Chingola, Zambia, as an example of holistic regenerative business models for mine closure. Without explicitly defining or operationalizing the concept of regeneration, Ref. [31] explains how contextual mechanisms were designed to promote positive connections among social, environmental, and economic systems, aligning with corporate goals (cost reduction, stability, and licensing). He stresses that identifying untapped resources and exploring their potential for synergy benefited local communities as well as the environment, while also creating profits, thereby fostering system resilience.

Turning our attention to food supply chains, Ref. [32] provides a more detailed perspective. They defined regenerative farming, quoting [33] as a farming approach that goes beyond mere sustainability. Regenerative farming leverages the natural regenerative tendencies of ecosystems when disturbed, and is characterized by practices that involve closed nutrient loops, reduced or eliminated use of biocidal chemicals, increased crop and biological diversity, a shift from annuals to perennials, and the emulation of natural ecological processes. They propose that regenerative business models can lead to "quadruple-aim performance" by fostering synergies among financial equity, ecological, human, and socioeconomic well-being.

In 2023, two more highly relevant overview articles were published [3,34]. While these two recent articles could not be integrated into our empirical study, they are included in the overview to enhance comprehension. The first [3] combines a review of the literature, anecdotal evidence, and focus group to investigate what regenerative business models are, and how organizations can create, deliver, and capture value in regenerative ways. They explain that business models of regeneration have barely been defined, and reviewed the existing literature on regenerative organizations with the aim to integrate them with the concept of business models.

Ref. [34] asserts that our understanding of regenerative systems at social-ecological scales remains limited, and presents a framework, the "Regenerative Lens," which emphasizes positive reinforcing cycles of well-being within and beyond these systems. The authors posit that this requires five key qualities: (1) an ecological worldview embodied in human action; (2) mutualism; (3) high diversity; (4) agency for humans and nonhumans to act regeneratively; and (5) continuous reflexivity.

Based on our review of the scant literature focusing specifically on regenerative business (models), it can be concluded that most of the existing literature lacks detailed definitions and theoretical elaboration, relying instead on anecdotal evidence and providing limited insights. However, the reviewed studies do showcase a diverse array of perspectives, encompassing ecological restoration, resource regeneration, and socio-ecological considerations. Scholars emphasize various aspects, including circular economy principles, eco-spirituality, and the imperative for a systems-based approach. These findings underscore the multifaceted nature of regenerative business models and highlight the urgent need for further research. Collaborative efforts are essential to establish clear definitions and key characteristics, advancing the theoretical understanding and empirical investigation of this nascent and complex field.

## 3. Methods

This study uses the Delphi Method, which is particularly suitable when existing research is limited [35,36]. The Delphi Method facilitates the inclusion of a diverse group of academic experts and practitioners, thus enabling the integration of multiple fields of expertise to collaboratively establish a definition of Regenerative Business Models. The Delphi Method involves a group decision-making process with participants possessing expertise in a relevant field [37]. Drawing inspiration from the Modified Delphi Method by [11], which builds on methods outlined by [38] and by [37], our study follows a seven-step approach to solicit opinions from experts and have them rank these opinions (see Figure 1).

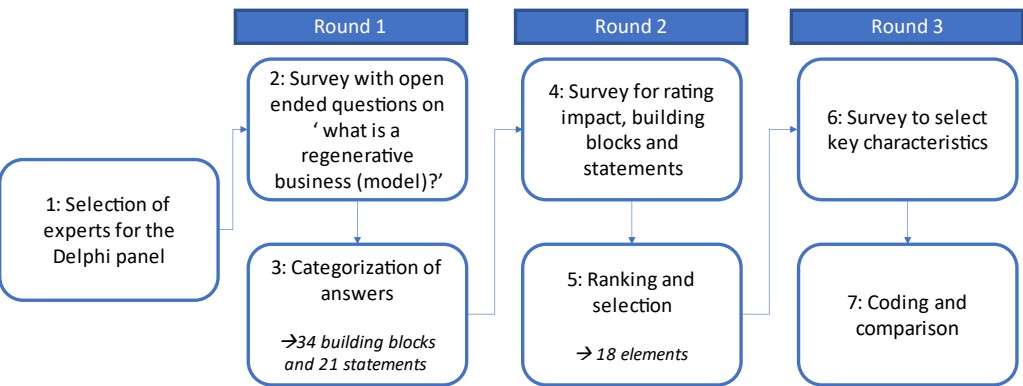

**Figure 1.** Summary of methodology.

### 3.1. Participants

A Delphi study builds on the expertise of its participants [37]. Expert selection criteria in emerging fields such as regenerative business models pose a challenge. In this study, we classified experts into three categories: academics, business representatives, and experts from the business eco-system, each with distinct selection criteria. Following the approach by Okoli, C. and S. D. Pawlowski [37], we identified potential experts within each category, starting with our own contacts and expanding the list through additional searches and nominations by other experts.

In their approach, Ref. [37] recommend including 10–18 experts on a panel. However, because three categories of experts are invited in our study, we invited a larger group, aiming for a representation of ten experts per group.

- Academics: Experts in this category were identified through a scientific literature review on Regenerative Business and the development of Sustainable Business Model concepts. Key authors were identified and approached through their institutional email as listed in recent publications.
- Business representatives: These experts include individuals associated with businesses that actively pursue sustainability and regeneration as part of their mission. This group consisted of entrepreneurs and CEOs only.
- Ecosystem experts: Experts in this category were nominated by the other experts. This category included opinion leaders, consultants, and policy makers that have recognizable expertise on regeneration. The policy makers have employment affiliations at local or national government departments. Opinion leaders and consultants have demonstrated experience through podcasts, documentaries, interviews, or other published media, or have received an award in the field.

The same people were invited for each round even if they had not responded on an earlier round. Only those indicating that they did not want to participate were no longer invited. Table 1 shows the number of surveys completed for each round.

**Table 1.** Survey completion per round.

|       | Round 1 | Round 2 | Round 3 |
|-------|---------|---------|---------|
| Total | 26      | 18      | 20      |

### 3.2. Data Collection and Analysis

The first round consisted of open-ended questions to identify key aspects of regenerative businesses and regenerative business models. Participants received an email invitation containing a link to an online survey in Qualtrics. A reminder was sent after ten days, and the survey was closed after twenty days. All three rounds were designed to be concise to minimize participants' time investment and to maximize response rates.

The first round contained three sections. The first section gathered information about participants' experience. The second section contained three open ended questions on the concept of regeneration: "What does the concept regenerative mean to you?", "What does regenerative business imply in your view?", and "What characterizes a regenerative business model in your opinion?". The third section included three questions to gather more detailed information on the concept of the regenerative business model: "What, in your opinion, are the key building blocks of a regenerative business model?", "What, if any, differentiates a regenerative business model from a circular business model in your view?", and "What, if any, differentiates a regenerative business model from a net-positive business model in your view?". At the end of the survey, participants were asked to list any company or person that they considered relevant for inclusion in this study, and an open field was included for additional remarks.

Content analysis of round one involved coding and categorization of responses using MaxQDA Version 3.0, which was conducted by the primary author. As an experiment, the second author employed ChatGPT version 3.0 to conduct an independent analysis. Specifically, the lists of answers given to each of the open-ended questions were fed separately, after which the system was requested to "identify the most relevant patterns from the list of answers and to indicate why they were considered most important". After that, the system was asked to compare the answers given regarding regenerative, 'sustainable', and 'net-positive'. Finally, the system was asked if it considered "the concept of regenerative business models to be sufficiently different from both sustainable business models and net-positive business models to constitute actual different concepts or whether the differences were only in the nuances" (which it confirmed). The authors jointly compared the categories derived from their own analysis and those generated by ChatGPT, which exhibited substantial overlap. These combined findings informed the design of the second round.

In the second round, after removing overlap, the answers were fed back to the participants as a list of 21 statements about regenerative business (RB: 9 statements) and Regenerative Business Models (RBM: 12 statements) on a four-point scale from Strongly disagree to Strongly agree. Statements included, for example: "RB has a model with a holistic approach, that looks at all its social and environmental impacts and makes sure to be fair, equal, replenishing what it uses" and "the regenerative values are embedded in the financial revenue model of the business, not externalized in charity or other external compensations".

Round two also involved asking experts whether they agreed that the identified list of impact areas represented the key elements of a regenerative business. Additionally, in round two, we included nine statements regarding the differences between Regenerative and Circular Business and ten statements regarding the differences between Regenerative and net-positive business. Respondents were asked to indicate their agreement with these statements on a four-point Likert scale ranging from Strongly disagree to Strongly Agree, and analyzed using descriptive statistics functions in SPPS.

Finally, in round three, experts were asked to confirm whether the list of the 18 highest ranking impact areas (from round two) represented the key elements of a regenerative business. Additional comments were collected from experts, including remarks about duplicates, redundancies, and the role of humans or social aspects. A follow-up question explored whether regenerative businesses should impact all identified fields or a selection.

## 4. Results

This section starts by outlining the similarities and differences that were identified when asking about regeneration, regenerative business and regenerative business models as concepts. This is followed by results that dive deeper into characteristics of regenerative business models. The final paragraphs of this section reflect on the difference of the concept of a regenerative business model in comparison to net-positive and circular models.

### 4.1. Regenerative, Regenerative Business, and Regenerative Business Models

Three open ended questions were included in the study to gain more insight in regenerative business models as a concept, asking about regenerative, regenerative business, and regenerative business models. Answers in relation to the concept regenerative emphasize that regenerative goes beyond the concept of sustainability. While sustainability focuses on maintaining the status quo and avoiding harm, regeneration involves actively restoring and replenishing natural and social systems. When comparing the answers about these three concepts, it is not surprising that there are many commonalities. The concepts of regenerative business and regenerative business models build on the regenerative perspective, but focus more on how businesses could contribute to this. Depending on which concept the experts reflect on, they do, not surprisingly, use different language and wording. Next to language, we identified three other topics for distinctions between the concepts: the scope and focus, the emphasis on value creation, and the approach to restoration and healing.

Table A1 in the appendix summarizes the main differences and commonalities between the concepts regenerative, regenerative business, and regenerative business models, using these four topics.

The findings show that both regenerative and regenerative business focus on making things better by restoring and positively impacting the world. However, when describing regenerative, the experts mainly focus on the environment and natural systems, while in describing regenerative business it goes further by applying these ideas to businesses, social systems, and other areas. The key point of regenerative business is that companies should actively work to bring positive changes and benefits to society, while also considering their impact on people, nature, and the economy.

Upon further exploration of the differences between regenerative business and regenerative business models, it becomes apparent that numerous experts explicitly view these concepts as synonymous. For instance, we received answers such as: "that was

my previous answer"; "same"; "see previous". Yet, based on the pattern of answers, we deduced several nuanced differences which can be summarized as follows:

**Scope and focus**: Whereas the experts relate regenerative business only to businesses, they relate the regenerative business model also to actors beyond the organizational level. For regenerative business, the experts emphasize how businesses can co-create, co-capture, and co-deliver economic, social, and environmental value while enriching the system they operate in. The focus is on businesses actively contributing to the restoration and well-being of social-ecological systems. In contrast, the concept of regenerative business models, according to the experts, does not only apply to businesses and the private sector, but equally beyond the organizational level: to communities, value chains, and social systems. When asked about regenerative business models, the experts reflect on how regenerative practices can positively impact various stakeholders and systems, encompassing both ecological and societal dimensions.

**Emphasis on Value Creation**: For both concepts the experts emphasize positive value, but for regenerative business models they are more explicit that this should encompass multiple forms of value creation. For regenerative business, they emphasize the importance of value creation and positive impact, with a specific focus on net-positive results. Going beyond "doing no harm" and aiming to maximize positive social and environmental outcomes are seen as key aspects. This also applies when addressing regenerative business models, but they are more explicit about the importance of broadening this to multiple forms of value creation, such as ecological value and value within communities and networks. Some experts also explain that regenerative business models build social and ecological capital through economic investments, ensuring that the entire value chain is positively influenced.

**Approach to Restoration and Healing**: Both in describing regenerative business and regenerative business models, restoration and healing are key aspects. However, the approach to how to contribute to this is described slightly differently. The experts link regenerative business to actively restoring and replenishing natural resources and ecosystems. They indicate such businesses aim to give back more than they take, and adopt circular economy paradigms. When reflecting on regenerative business models, they furthermore address the need for a holistic approach to regeneration. Regenerative business models, according to the experts, are characterized by a well-developed, transparent plan to minimize negative impacts across all systems, considering potential side-effects on other systems.

In summary, according to the experts, regenerative business models highlight businesses' role in co-creating value, positive impact, and ecological restoration. Regenerative business models are seen as encompassing various contexts beyond businesses and emphasizing multiple forms of value creation, social-ecological system restoration, and the need for a holistic approach to regeneration.

### 4.2. Characteristics of Regenerative Business Models

The previous section showed a comparison between the expert opinion on the general concepts regenerative, regenerative business, and regenerative business model. This section delves deeper into the aspects defining regenerative business, and specifically regenerative business models.

#### 4.2.1. Impact Areas for Regenerative Business

In round one, we identified a list of impact areas. In round two, we presented these and asked the experts to indicate how important they considered them to be, as potential parts of a definition (on a 5-point Likert scale). Based on average scores, all impact areas were considered to be at least 'important'. Working conditions scored lowest with an average of 3.70 on a 5-point Likert scale, and biodiversity scored highest with an average of 4.91. To determine consensus, we identified the percentage of experts scoring the area as very or extremely important and only included those areas scoring 75% or more, resulting

in a list of six impact areas. The list consisted of: Biodiversity; Community; Environment; Humans; Nature; Social-Ecological. Furthermore, we noticed considerable agreement on the importance of biodiversity and nature, while more than 95% considered these very important, over 80% of the respondents even indicated that these aspects are extremely important. In round three, we asked the experts to indicate whether they agreed that the list (marked grey in Table 2) represents the key elements upon which a regenerative business impact. A small majority of 14 respondents answered affirmatively.

**Table 2.** Rated importance for area of impact.

| Impact Area | Mean Value | % Above Very Important |
|---|---|---|
| Biodiversity | 4.91 | 100% |
| Nature | 4.87 | 95% |
| Environment | 4.43 | 90% |
| Communities | 4.35 | 75% |
| Social-ecological system | 4.35 | 75% |
| Humans | 4.04 | 75% |
| Health (of people) | 4.04 | 65% |
| $CO_2$ reduction | 3.78 | 65% |
| Local economy | 3.87 | 55% |
| Knowledge | 3.74 | 55% |
| Working conditions | 3.70 | 45% |

As a follow-up question, we asked the respondents whether regenerative business should impact each of these fields (instead of a selection). This resulted in a mixed bag, with 10 experts indicating this should be the case and another 10 indicating regenerative businesses could also impact one or more of these fields.

4.2.2. Building Blocks of Regenerative Business Models

In round one, we asked the respondents to indicate (maximum three) key building blocks of regenerative business models, which yielded 34 building blocks in total. In the second round, we presented these 34 building blocks, in their own wordings. We asked which of these building blocks they considered vital, without indicating how many they could/should select. On average, the experts selected 8 building blocks, with a range of 4 to 13 building blocks being selected. Table 3 lists all building blocks, sorted by the number of times selected.

Four building blocks were selected by at least half of the experts (nine or more). These were: 'Deep understanding of impacts and dependencies on nature, people, and society', 'Desire to keep learning and adapting based on the business' activities in larger wholes (natural, social, and human systems)', 'Collaboration and care over competition', and 'Positive impact'. The majority of experts selecting these implies that they are widely acknowledged as essential elements. Four building blocks were not selected at all.

To determine their relative importance, we also asked the respondents to rank the selected building blocks, with 1 for the most important building block, 2 for the next, and so on. Table 3 shows that 12 different building blocks were selected as being most important. Additionally, for some building blocks, there is a large variation in ranking. For instance, 'the Desire to keep learning and adapting...' was ranked both 1st and 10th, with an average rank of 5th. 'Holistic' has the lowest average rank. This building block was only selected twice, but both times considered most important. As such, it can be concluded that, when it comes to the building blocks of regenerative business, little consensus exists amongst the experts.

**Table 3.** Building blocks ranked by importance.

| | Times Selected | Highest Rank | Lowest Rank | Mean Value |
|---|---|---|---|---|
| Deep understanding of impacts and dependencies on nature, people, and society | 15 | 1 | 11 | 3.47 |
| Desire to keep learning and adapting based on the business' activities in larger wholes (natural, social, and human systems) | 11 | 1 | 10 | 5.27 |
| Collaboration and care over competition | 9 | 3 | 9 | 6.22 |
| Positive impact | 9 | 1 | 12 | 3.56 |
| Embedding purpose in decision making and corporate structure | 8 | 1 | 10 | 5.25 |
| Long term approach (20–100 years) | 8 | 1 | 8 | 4.25 |
| Holistic, inclusive, and diverse regenerative leadership | 7 | 5 | 9 | 6.14 |
| Earth systems-view | 7 | 1 | 4 | 2.57 |
| Stakeholders (human and nonhuman) | 6 | 1 | 9 | 4.67 |
| True pricing | 6 | 1 | 7 | 4.17 |
| Co-creation | 5 | 2 | 10 | 5.60 |
| Inner (conscious) and outer learning and development | 5 | 3 | 9 | 5.60 |
| No harmful emissions (GHG nor pollutants) | 5 | 3 | 10 | 5.60 |
| Multiple value proposition | 5 | 3 | 9 | 5.40 |
| Viable, desirable, and feasible business | 4 | 5 | 13 | 7.75 |
| Radical collaboration | 4 | 2 | 8 | 5.75 |
| Circularity | 4 | 3 | 12 | 5.75 |
| Good for all stakeholders | 4 | 1 | 6 | 3.50 |
| Generous by design | 3 | 3 | 11 | 6.00 |
| Sustainable supply chain | 3 | 1 | 9 | 5.67 |
| Closed material loops | 3 | 1 | 5 | 3.00 |
| Trustworthy government | 2 | 9 | 11 | 10.00 |
| Fair prices in the supply chain | 2 | 5 | 8 | 6.50 |
| Part of a network | 2 | 4 | 8 | 6.00 |
| Multiple returns | 2 | 2 | 8 | 5.00 |
| Holistic | 2 | 1 | 1 | 1.00 |
| Social Flows | 1 | 10 | 10 | 10.00 |
| (Holistic) knowledge by managers and employees | 1 | 8 | 8 | 8.00 |
| Avoidance of inputs withdrawing from nature | 1 | 5 | 5 | 5.00 |
| Personnel development | 1 | 3 | 3 | 3.00 |
| Energy flows | 0 | | | |
| Equality | 0 | | | |
| Use of unused energy | 0 | | | |
| Transformation of environmental negative inputs (waste) into secondary raw materials | 0 | | | |

4.2.3. Statements on Regenerative Business and Regenerative Business Models

To delve deeper into the concepts of regenerative business and regenerative business models, we presented the experts in the second round a series of 21 statements, based on the descriptions and definitions they provided in round one. Nine statements were

categorized as statements about regenerative business and twelve more specifically about a regenerative business model. These were statements relating to how regenerative is embedded in the business. The results are shown in Tables 4 and 5.

**Table 4.** The nature of regenerative business.

| Regenerative Business | Min. | Max. | Mean | Std. Dev. |
|---|---|---|---|---|
| RB focuses not only on the preservation and sustainable and conscious consumption of resources, but also on their renewal and restoration | 3 | 4 | 3.80 | 0.410 |
| The positive impacts on nature, people, and society are embedded in decision making and purpose | 3 | 4 | 3.65 | 0.489 |
| RB has a model with a holistic approach, that looks at all its social and environmental impacts and makes sure to be fair, equal, replenishing what it uses. | 2 | 4 | 3.50 | 0.607 |
| RB means going significantly beyond "doing no harm" and delivering positive value even if there are no direct incentives | 1 | 4 | 3.50 | 0.761 |
| In RB all aspects of the business strategy cover economic, social and environmental value and external resources (bio stocks, ecosystem services, and actors) are included as if they were their own | 2 | 4 | 3.40 | 0.598 |
| RB makes a net-positive impact on society with regard to material use | 2 | 4 | 3.40 | 0.681 |
| A regenerative business is a business that has the capacity to innovate to recover from a difficult challenge | 1 | 4 | 2.55 | 0.945 |
| RB means earning your money with cleaning up pollution, capturing CO2, planting trees, helping others to become more sustainable, etc. | 1 | 4 | 2.50 | 1.051 |
| There is no difference between what characterizes a regenerative business or what characterizes a regenerative business model | 1 | 4 | 2.35 | 0.875 |

The findings suggest that (based on the lower standard deviations) there is slightly more consensus on what constitutes regenerative business compared to what constitutes a regenerative business model.

In round one, some of the experts added a comment that both concepts are the same. Therefore, it is noteworthy that no consensus was observed about whether there is a difference in what characterizes a regenerative business and what characterizes a regenerative business model.

There is, however, consensus that regenerative business focuses on renewal and restoration, delivers positive value, looks at all social and environmental impacts, and that the positive impacts are embedded in the business. Regarding regenerative business models, there is some consensus that they use their influence to inspire stakeholders, communities, and everyone in the value chain, and it focuses on multiple forms of value creation, embeds regenerative values in the financial revenue model, and builds long-term customer relations.

**Table 5.** The nature of regenerative business models.

| A Regenerative Business Model | Min. | Max. | Mean | Std. Dev. |
|---|---|---|---|---|
| It has sustainability not just at the office but at the core business and uses its influence to inspire stakeholders to join | 2 | 4 | 3.50 | 0.607 |
| It has a focus on multiple forms of value creation, with an explicit link to ecological value and a net-positive result | 2 | 4 | 3.40 | 0.681 |
| It positively influences everyone in the value chain and the communities touched through this chain, including the culture of the organization | 2 | 4 | 3.30 | 0.657 |
| It is based on long term value and a two-way relationship with the customer | 1 | 4 | 3.25 | 0.716 |
| It generates four returns: natural return, return of inspiration, social return, and financial return (model by Commonland) | 2 | 4 | 3.20 | 0.616 |
| the regenerative values are embedded in the financial revenue model of the business, not externalized in charity or other external compensations | 2 | 4 | 3.05 | 0.759 |
| Companies must measure their process under the material and energetic perspective, identifying the main hotspots and the main economic, environmental, and societal implications | 1 | 4 | 3.00 | 0.858 |
| It builds on a purpose case, creating value in networks between organizations | 2 | 4 | 2.95 | 0.686 |
| It builds social and ecological capital through the investment of economic capital | 1 | 4 | 2.95 | 0.759 |
| It has a well-developed, transparent plan to minimize negative impacts across all systems | 1 | 4 | 2.85 | 0.875 |
| It is dependent on all stakeholders, a trustworthy government, and regulatory authorities | 1 | 4 | 2.70 | 0.979 |
| It requires valuation of human capital and of nature in money, time, use, etc. | 1 | 4 | 2.65 | 1.089 |

The combined responses about area of impact, building blocks, and the statements show the importance of both natural and human aspects as part of the definition of regenerative. The building blocks show that interdependencies should be understood and that businesses should be considered as part of the greater whole (natural and human systems). Comparing regenerative business with regenerative business model, this again emphasizes the positive influence of both, mentioning social and environmental impacts. It also underlines the notion that when speaking about regenerative business model, more emphasis is put on how it influences stakeholders outside the organization (inspire stakeholders, communities, and everyone in the value chain) and up to some level the integration into business. It is noteworthy, however, that how it is integrated is described in abstract terms, which may partly result from the research approach. In addition, from answers to the statements, we might infer that regenerative businesses and business models are characterized by intentional and proactive agency in their commitment to create positive impacts, and that sustainability can be considered as an integral component of their operations and ethos.

### 4.3. Regenerative versus Circular and Net-Positive Business Models

In addition to gaining insights into the concept, we also wanted to explore how it differs from related concepts. Hence, for the first round, we asked the experts to reflect on the potential differences between a regenerative business model and two related concepts, namely, circular business models and net-positive business models. Tables 6 and 7 provide an overview of listed differences.

**Table 6.** Comparison of regenerative and circular business models.

| | Regenerative Business Models | Circular Business Models |
|---|---|---|
| Value Creation and Focus: | Goes beyond resource efficiency. Focus on actively increasing the value of the environment and aim to restore and renew natural and social systems and seek to have a positive impact on ecosystems and communities. | Focus on "closing the loop" and aim to reuse, reduce, and recycle resources. The primary concern is to prevent value loss from the environment by efficiently managing material flows. |
| Holistic Approach and Social Considerations | Adopt a more holistic approach, encompassing not only material flows, but also social, ecological, and human systems. They consider the overall well-being of the environment and society in their decision-making. | Primarily concentrate on resource flows and material efficiency. They may not always consider the broader social and ecological impacts of their operations. |
| Sustainability vs. Restoration | Go beyond sustainability and focus on restoration. They aim to actively regenerate natural resources and ecosystems, going beyond just preventing depletion. | Often focus on sustainability, aiming to stabilize resource flows and prevent the depletion of resources. They seek to maintain the current status quo and minimize negative impacts. |
| Nature Positive and Surplus Generation | Are associated with being "nature positive" and generating more positive impact than required. They are about giving back to nature and communities, creating a surplus of value. | May not necessarily generate a surplus of positive impact. While they aim to close loops and be sustainable, they may not always produce additional benefits for nature and society. |
| Relationship with Customers and Purpose | In addition to focusing on customers, take a broader perspective by incorporating environmental, social, and community considerations in their purpose-driven approach. | May focus on efficient resource use and value delivery to customers. Their emphasis is on the relationship with customers and delivering value through resource management. |

When analyzing the expert's answers, five potential nuanced differences between regenerative and circular models were identified, as shown in Table 6. The differences between regenerative and circular models relate mainly to goals, scope, and focus, and they illustrate that regenerative and circular business models are distinct concepts, representing different approaches to sustainable business practices. While they both share a commitment to sustainability and environmental consciousness, their core principles and methodologies appear to set them apart as separate and unique concepts.

When exploring the similarities and differences between regenerative business models and net-positive models, we found that, according to some experts, considerable overlap exists, with a net-positive model potentially comprising both regenerative and circular elements, indicating that there may be variations and interpretations of each model depending on context and application. Yet, the overall patterns of answering suggest that regenerative and net-positive business models are distinct concepts, with the former emphasizing restoration, positive impacts, and a holistic approach, while the latter may have a narrower focus on specific measurable aspects such as emissions and carbon impact. More specific differences are summarized in Table 7.

For round two, we presented the answers regarding the differences between regenerative business models and circular and net-positive business models into two sets of

statements. We asked the participants to indicate their agreement on these statements, and the results are shown in Figures 2 and 3.

**Table 7.** Comparison of regenerative and net-positive business models.

| | Regenerative Business Models | Net-Positive Models |
|---|---|---|
| Earth as the Point of Departure for Regenerative | Are often associated with being centered around the Earth or the environment as the starting point. They aim to restore and regenerate natural systems and ecosystems | |
| Scope of Application | Some answers suggest that these are specifically related to agriculture. | Have a broader application beyond agriculture. |
| Positive Impact vs. Compensation | Focus on creating positive impacts on the environment, communities, and ecosystems. They go beyond just compensating for negative impacts and actively contribute to restoration and well-being. | May focus on plussing and minning, striving for positive impacts while also acknowledging the possibility of negative impacts. However, some lines mention that "net-positive" models allow for negative impacts, whereas regenerative models exclude a damage model. |
| Measurement and Inclusion of Non-Directly Measurable Aspects | Mentioned to include non-directly measurable aspects, suggesting a more holistic approach to impact assessment. | May be more focused on measurable aspects, particularly related to emissions and environmental indicators. |
| $CO_2$ vs. Nature Positive | | Are associated with being $CO_2$-positive, mainly focusing on emissions and carbon-related impacts. |

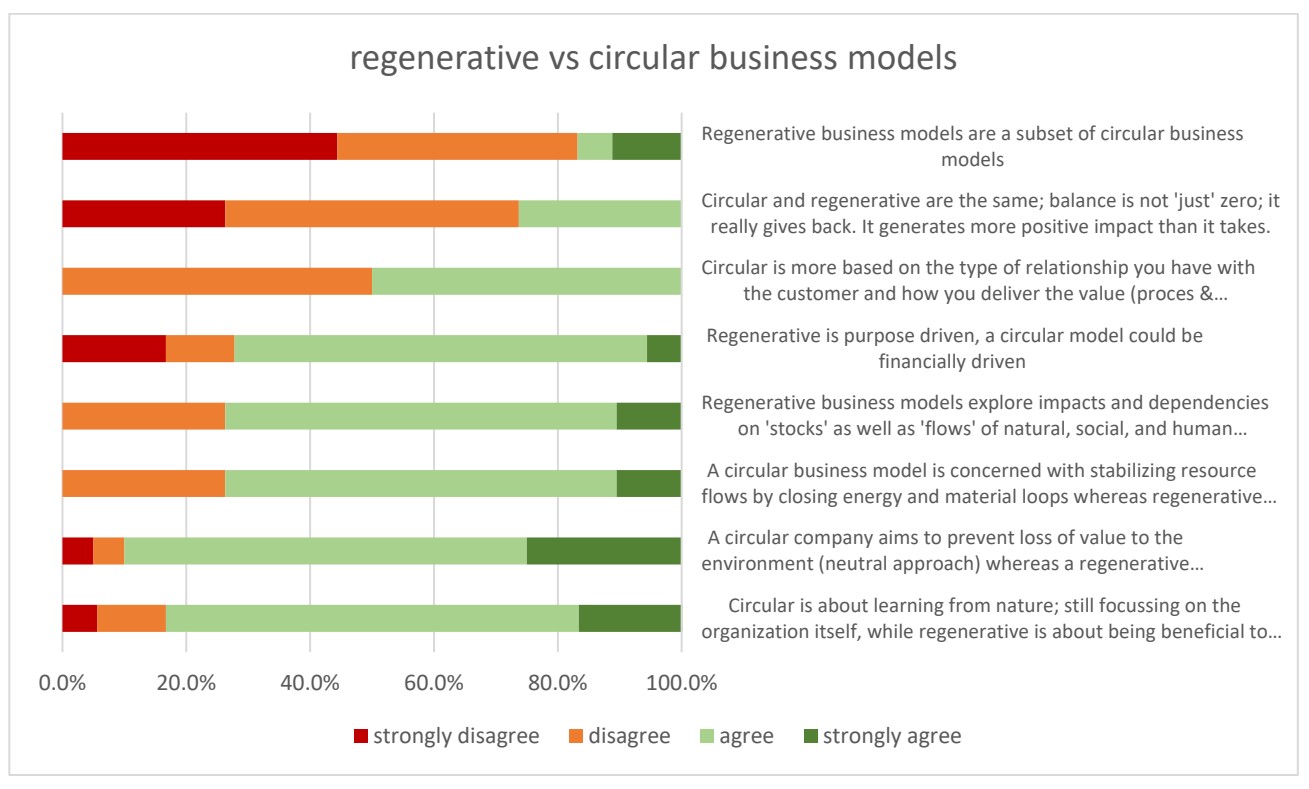

**Figure 2.** Regenerative and circular business models compared.

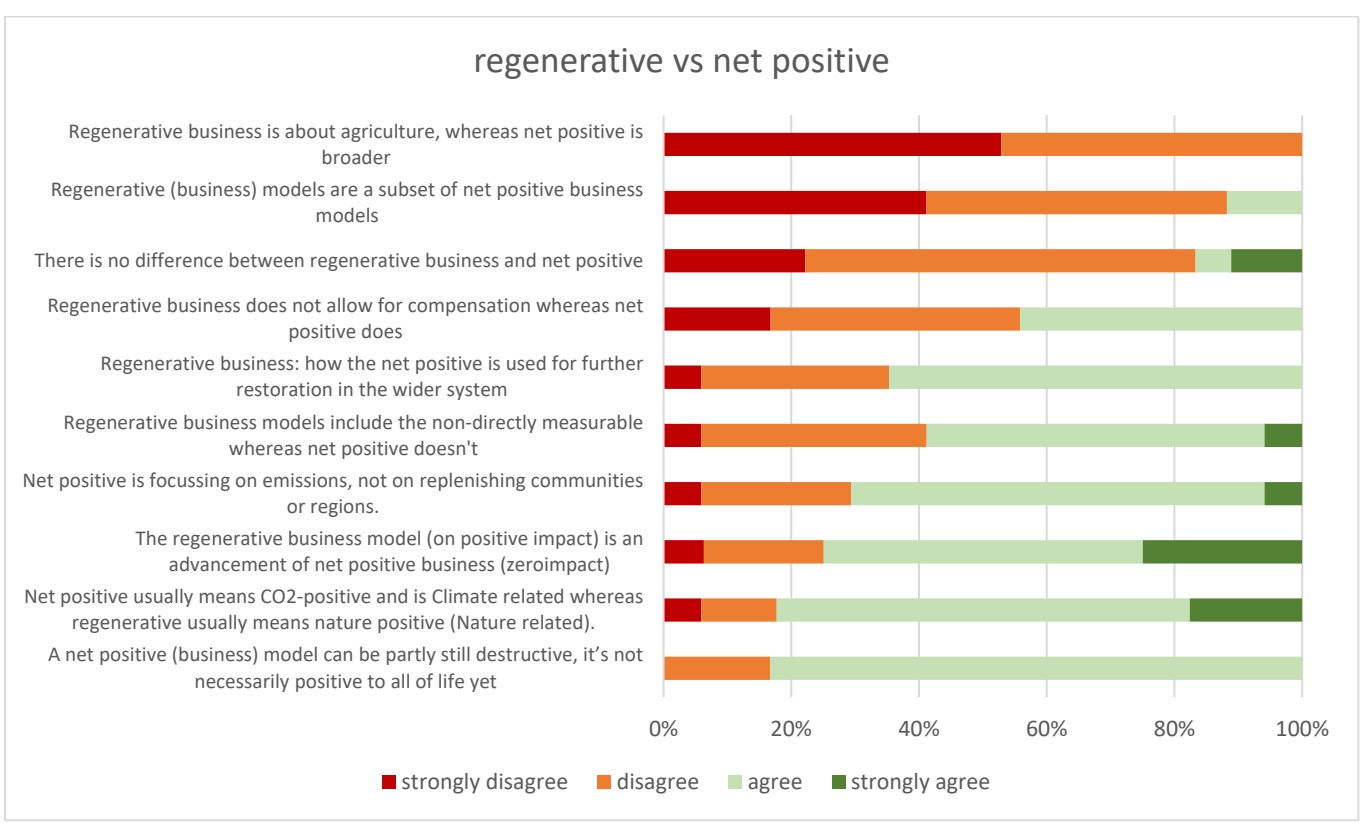

**Figure 3.** Comparison of regenerative vs. net-positive business models.

These results show that participants generally disagree with the statements that regenerative business models are the same or a subset of either circular or net-positive models, supporting the notion that regenerative business models are a separate category that deserves attention. In addition, all participants disagree with the notion that regenerative business is about agriculture, whereas net-positive is broader (100%).

There is consensus (with more than 80% either (strongly) agreeing or (strongly) disagreeing) on the statement that circular models aim to prevent loss, whereas regenerative models increase value, meaning that regenerative models have a positive instead of a neutral approach (90%). Another difference on which there is consensus is that circular focuses on the organization itself, whereas regenerative is about beneficial to all: functioning as nature (83.4%).

There is consensus about two statements regarding differences between net-positive and regenerative models, being: a net-positive (business) model can be partly still destructive, it is not necessarily positive to all of life yet (83.3%) and net-positive usually means $CO_2$-positive and is climate-related, whereas regenerative usually means nature positive (Nature related) (82.3%).

Areas where such consensus is currently lacking include:

- Circular being based on customer relationships and value delivery vs. regenerative being based on the more holistic approach to how you conduct business (50–50%);
- Regenerative business models include the nondirectly measurable whereas net-positive does not (41.2–58.8%);
- Regenerative business does not allow for compensation whereas net-positive does (55.6–44.4%).

## 5. Discussion

The escalating global urgency to adopt sustainable business practices, fueled by environmental and social challenges, has given rise to an array of emerging business practices and models. Our study addresses the current gap in understanding these models by delv-

ing into the evolving field of regenerative business models. In employing a Delphi-inspired approach [37], our study incorporates perspectives from academic experts, business representatives, and professionals in the broader business ecosystem. Through this approach, we seek to contribute to the knowledge on regenerative business models, specifically by exploring key design principles aimed at discerning the unique characteristics that set regenerative business models apart from other sustainable paradigms. This study aimed to answer two research questions: What are regenerative business models? And how do they differ from other sustainable business models, specifically circular and net-positive models?

In response to the first question, the findings reveal that regenerative, regenerative business, and regenerative business models share the overarching goal of fostering positive change and impact on the world. However, nuances among these concepts emerge, suggesting distinctions in their scales of application and contextual relevance. Organizations adopting regenerative business models focus on planetary health and societal well-being. They generate value across multiple stakeholder levels, including nature, societies, customers, suppliers, partners, shareholders, investors, and employees. This is achieved through activities promoting regenerative leadership, co-creative partnerships with nature, and justice and fairness. These organizations aim for a net-positive impact across all stakeholder levels by capturing value through multicapital accounting.

Our study underscores the significance of a holistic approach and systems thinking in the context of regenerative business, aligning with contemporary approaches to sustainability [3,39]. This emphasis on interconnectedness highlights the need for entrepreneurs to consider their profound effects on nature, communities, and society. Regenerative business models propose the new goal and framing of planetary health and societal wellbeing, and advance a motivating narrative that aims not merely at the reduction of negative impacts (net zero) or the balance between economic, social, and environmental value creation (triple bottom line) [22], but at a redefinition of the relationship between humans and nature. Noteworthy is that while experts emphasize environmental and social aspects in their responses, economic dimensions of business models such as the nature of the products and services offered, costs and revenue structures, marketing, and logistics [22] received minimal attention. This observation raises questions about whether a regenerative business model is perceived more as a network or ecosystem concept than a model tailored for individual businesses, a viewpoint that contrasts with previous research emphasizing concrete regenerative practices for small- and medium-sized enterprises [4].

Addressing the second question, the findings suggest that regenerative models, circular models, and net-positive models represent distinct approaches to sustainable business practices. According to the experts in this study, regenerative business models extend beyond mere resource efficiency, actively enhancing environmental value by restoring and renewing natural and social systems. In contrast, circular models concentrate on "closing the loop," emphasizing the efficient management of material flows to prevent value loss from the environment.

Exploring the relationship between regenerative business models and net-positive models, we discovered considerable overlaps, with some experts suggesting that a net-positive model could encompass regenerative and circular elements. However, a tentative consensus emerged, suggesting that regenerative models emphasize restoration, positive impacts, and a holistic approach, while net-positive models may have a narrower focus on measurable aspects such as emissions.

Comparing and contrasting the concepts of the regenerative business model with circular and net-positive business models allowed us to illuminate the unique features and contributions of each model, thereby fostering a nuanced understanding of their distinct characteristics and contributing to the future development of a comprehensive framework for sustainable and regenerative practices. Additionally, contrasting these models will enable entrepreneurs to engage in informed decision-making regarding their sustainability goals and practices. In this study we only compared regenerative business models to net-positive and circular models, whereas future research might benefit from comparison with

a broader set of concepts. Other partially overlapping models and labels are, for instance, eco-positive [40], restorative, and reconciliatory and positive sustainability [41,42].

Despite the valuable insights gained, it is essential to acknowledge the limitations of our Delphi-inspired approach. The number of participants was limited and during the study, we noticed a decline in participation of practitioners. When asked, they indicated that there was too much emphasis on defining the concept, therefore they couldn't relate to the questions asked. For future studies, the experts' group may be separated in analysis. In addition, the approach mostly focused on consensus and not on disagreements. While experts agreed on some aspects, in many areas consensus was lacking, thus emphasizing the need for continued exploration and refinement of these concepts in both research and practical application. Future research should address these limitations by refining the methodology and actively exploring disagreements to foster a more genuine consensus.

In conclusion, our findings underscore a key theoretical contribution—moving beyond anecdotal evidence to establish a systematic and theoretically grounded understanding of regenerative business models. However, this study only serves as a starting point, acting as a catalyst for further theoretical and empirical exploration. It encourages ongoing debate and research in this crucial domain, bridging the gap in the existing literature.

**Author Contributions:** L.D. and I.W. have designed and analyzed the Delphi study together and have written equal parts of this paper, by discussing key points for each section and writing iteratively. Selection and approach of participants was performed by the first author. All authors have read and agreed to the published version of the manuscript.

**Funding:** This Research received no external funding and was supported by the Centre of Economic Transformation of Amsterdam University of Applied Sciences.

**Institutional Review Board Statement:** Not applicable.

**Informed Consent Statement:** Informed consent was obtained from all subjects involved in the study as part of the survey.

**Data Availability Statement:** The data presented in this study are available on request from the corresponding author (accurately indicate status).

**Acknowledgments:** We would like to thank the expert participants to our Delphi for sharing their views during the (up-to) three rounds of our study. Additionally, we like to thank the participants in the 2023 NBM Conference in Maastricht for their valuable comments about an earlier version of this work.

**Conflicts of Interest:** The authors declare no conflicts of interest.

## Appendix A

**Table A1.** Comparative patterns for regenerative, regenerative business and regenerative business model.

| | Regenerative | Regenerative Business | Regenerative Business Models |
|---|---|---|---|
| Description | Experts emphasize the transformative and positive nature of regeneration, its focus on healing and restoring ecosystems and the importance of integrating with nature to achieve long-term sustainability and well-being. | Experts focus on businesses and their potential to contribute positively to the well-being of the environment and society. Answers underscore the need for businesses to consider their dependencies and impacts on all ecological and societal aspects | Experts highlight the value-driven, restorative nature of regenerative business models and underscore the importance of considering both ecological and economic aspects in a holistic and balanced approach. |

**Table A1.** *Cont.*

| | Regenerative | Regenerative Business | Regenerative Business Models |
|---|---|---|---|
| Language and orientation | • A general and diverse set of words<br>• Personal views and definitions<br>• Focus on the intrinsic nature of sustainability | • Business-oriented words<br>• Specific mentions of business models, strategy, and the role of the private sector<br>• Reference to circular paradigms, economic impacts and the role of innovation | • Business-oriented words<br>• Specific mentions of business models, strategy, and the role of the private sector<br>• Reference to circular paradigms, economic impacts and the role of innovation |
| Scope and Focus | • Practices<br>• Natural systems, ecosystems and the environment<br>• Emphasizing biodiversity, ecosystem health, and alignment with natural processes.<br>• Positive feedback loops | • Business Approach<br>• Businesses, social systems, education, and innovation<br>• Emphasizing interconnectedness of social, environmental, and economic aspects<br>• Need for understanding of impacts and dependencies<br>• Co-creating value | • Innovating Systems<br>• Beyond organizational level, including communities, value chains, and social systems.<br>• Emphasizing balancing ecology and economy within the business model<br>• Co-creating value |
| Emphasis on Value Creation | • Understanding and working in harmony with nature<br>• Approach that considers ecological, social, and economic aspects, with a focus on creating circular resource flows.<br><br>Highlights the interconnectedness of all living systems. | • (Net) Positive Value Creation and Positive Impact on society, the environment and communities<br><br>Goes beyond "doing no harm" | • Multiple forms of value<br>• Net-positive impact on society and the environment.<br>• Prioritize positive social and environmental outcomes over mere profit and growth. |
| Approach to Restoration and Healing | Restoration<br>• Aims to heal and restore damaged ecosystems and social systems.<br>• Involves repairing and revitalizing what has been depleted, damaged, or lost.<br>• The emphasis is on regeneration as a process of renewal and rejuvenation after an accident or disturbance.<br>• "Recovery as the starting point of everything you do."<br>• "Actions do not deplete resources but rather ensure their regrowth and/or strengthening" | Flourishing<br>• Restoring and replenishing resources and ecosystems.<br>• Giving back more than they take and actively working towards repairing damages and creating value through a circular economy approach.<br>• Actively contributing to the well-being of the ecosystems and societies they operate in.<br>• The emphasis is more on the results and the approach, co-creation is a major element in this approach.<br>• "Allows for ecosystems to flourish and regrow through their activities" "it adds life to our lives rather than is extractive" | Holistic<br>• Designed to regenerate natural, human, and social systems while considering potential negative side-effects on other systems<br>• Holistic approach, recognizing the interconnectedness of various systems and the need to minimize negative impacts across all aspects of the business.<br>• "Enables a specific natural or societal system to restore, heal and thrive whilst overall not generating any negative impact on nature and society" |

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
