# Peer review of "Exploring Characteristics of Regenerative Business Models through a Delphi-Inspired Approach"

_sustainability, doi:10.3390/su16073062_

Round 1

Reviewer 1 Report

Comments and Suggestions for Authors

The manuscript presents itself with commendable clarity in its writing; however, it falls short in its discussion section, which is notably absent. This section is crucial for interpreting the findings in the context of existing literature, enabling a comparison between the current results and previous studies. Such a comparison not only validates the research findings but also situates the study within the broader academic conversation.

Furthermore, the article lacks a conclusive segment that should encapsulate the theoretical contributions, managerial implications, and the overarching significance of the research. These elements are essential for articulating the value of the study to both the academic community and practitioners in the field, offering insights that could inform future research, policy-making, or managerial practices.

Another significant omission is the discussion on the limitations of the research. Acknowledging the study's limitations not only enhances its credibility but also guides future research directions by identifying areas for improvement or further exploration.

To strengthen the manuscript, I recommend that the authors:

  • - Incorporate a detailed discussion section that compares the current findings with those of previous studies, highlighting both consistencies and discrepancies.
  • - Clearly articulate the conclusions, theoretical contributions, and managerial implications of the study to demonstrate its relevance and value to the field.
  • - Identify and discuss the limitations of the current research, providing a pathway for future scholarly inquiry.

Author Response

Dear reviewer,

Thank you for your comments and the well-supported feedback. In this new version, we added a discussion section, including comparison with literature (comment 1), a conclusive segment (comment 2) and limitations of the research (comment 3).

Reviewer 1

1.       it falls short in its discussion section, which is notably absent. I recommend to Incorporate a detailed discussion section that compares the current findings with those of previous studies, highlighting both consistencies and discrepancies

2.       the article lacks a conclusive segment that should encapsulate the theoretical contributions, managerial implications, and the overarching significance of the research. I recommend to clearly articulate the conclusions, theoretical contributions, and managerial implications of the study to demonstrate its relevance and value to the field.

3.       Another significant omission is the discussion on the limitations of the research. I recommend to Identify and discuss the limitations of the current research, providing a pathway for future scholarly inquiry.

Reviewer 2 Report

Comments and Suggestions for Authors

The study 'Exploring characteristics of Regenerative Business Models through a Delphi-inspired approach' addresses an interesting research topic worthy of scientific recognition. The authors focused their attention on defining regenerative business models and analysing their essence. By pointing out the importance of regenerative models in the process of environmental restoration and societal well-being, they emphasise the significance of the research strand undertaken. 

The abstract of the article presents the general subject of the research, the research gap in response to which the study was written, and outlines very generally the results of the findings. I believe that the results of the findings and their significance should be discussed more fully, and the reference to the essence of the Delphi method should be discussed in the section discussing research methods, in a separate section of the article. In this respect, the summary should be refined.

The introductory section presents the general background of the research, identifies the existing research gap and justifies the need for the research. This section outlines the direction and layout of the research. The stages of the research are worth capturing in the research model. I believe that the presentation of the general findings indicated in this section should be removed and included in summary form. The introductory section should be improved in this respect.

Section 2 'Theoretical Background' needs to be strengthened in terms of the literature review with regard to the background of the research. In this respect, it is worth studying https://doi.org/10.3390/su15118889, https://doi.org/10.3390/su141811695, among others. The literature of 36 items is very modest - this scope needs to be further developed.

Section 3 discusses the methodology adopted in the model.  The section needs to be reworked. It is worth discussing the methodology in coupling with the research steps adopted in the study (research model missing). In addition, the figure should be moved to line level 225 ( text with discussion is required after the figure, table).

The results are presented in sections, discussing each strand of the research, to ensure clarity of presentation. Under the table - line 434 should be completed with text. Similarly, line 458, 537.

The article lacks a "discussion" section, where the authors relate the results of the findings to the existing literature. Also missing is a 'summary' section where the authors summarise the findings, discussing their contribution to theory and practice. These elements absolutely need to be completed.

The literature, as already indicated in the review, needs strengthening. References should be developed according to MDPI standards.

In conclusion, the article tackles an interesting topic, but needs refinement. The base is good. The structure and scope of the content presented should be rethought and refined - comments to this effect are identified in the review.  In developing the 'discussion' section, attention should be paid to implications. When developing the 'summary' section, the focus should be on the findings in relation to the purpose of the research. The practical significance of the findings should be indicated, the novelty of the study should be pointed out.

Author Response

Dear reviewer,

Thank you for your timely and elaborate review. We have read the suggestions provided and greatly value your comments. We have aimed to improve our manuscript by adding a discussion section and including several clarifications to the main text.

Please find attached a specification of the provided comments and our adjustments to the manuscript.

Reviewer 3 Report

Comments and Suggestions for Authors

I consider the article “Exploring characteristics of Regenerative Business Models through a Delphi-inspired approach” suitable for publication.

The topic is highly current and the article contributes to the development of knowledge in the field of regenerative business models.

In addition, I consider the article to be very well prepared, with a high degree of comprehensibility and clearly stated conclusions.

My comments are rather partial and have the character of recommendations. They include:

1) Specify the terms "circular model" and "net-positive model" in more detail in the introduction or theoretical background (e.g. based on currently available literature).

2) Specify the composition of the group of respondents listed as "Business representatives" - which experts (managers) are they? What job positions do they work in? The other two groups of respondents are much more specifically defined.

3) Add research limitations and suggestions for further research (not only is this a standard part of scientific articles, but the authors also promise it in the introduction).

4) Check table numbering and related table references. In the text there is, for example, a link to tab. 8, but it does not exist.

Author Response

Dear reviewer,

Thank you your review and the kind suggestions for improvement, we greatly value your comments. We have aimed to improve our manuscript by adding a discussion section and including several clarifications to the main text.

Please find below a specification of the provided comments and our adjustments to the manuscript.

Reviewer 3

1.       Specify the terms "circular model" and "net-positive model" in more detail in the introduction or theoretical background (e.g. based on currently available literature).

Thank you for your suggestion. We added a clarification on these concepts in line 38

2.       Specify the composition of the group of respondents listed as "Business representatives" - which experts (managers) are they? What job positions do they work in? The other two groups of respondents are much more specifically defined.

We added a very brief comment in line 253 for clarification.

3.       Add research limitations and suggestions for further research (not only is this a standard part of scientific articles, but the authors also promise it in the introduction).

In this new version, we added a discussion section, including comparison with literature, a conclusive segment and limitations of the research.

4.       Check table numbering and related table references. In the text there is, for example, a link to tab. 8, but it does not exist.

Thank you for pointing this out, we checked all table numbers and adjusted them where necessary.

Round 2

Reviewer 1 Report

Comments and Suggestions for Authors

Congratulations to the author for the additions made to the article.

Reviewer 2 Report

Comments and Suggestions for Authors

The authors have taken on board the suggestions made in the review and improved the article.